# Systematic Analysis and Functional Characterization of R2R3-MYB Genes in *Scutellaria baicalensis* Georgi

**DOI:** 10.3390/ijms23169342

**Published:** 2022-08-19

**Authors:** Wentao Wang, Suying Hu, Caijuan Zhang, Jing Yang, Tong Zhang, Donghao Wang, Xiaoyan Cao, Zhezhi Wang

**Affiliations:** 1National Engineering Laboratory for Resource Development of Endangered Crude Drugs in Northwest China, Key Laboratory of the Ministry of Education for Medicinal Resources and Natural Pharmaceutical Chemistry, Shaanxi Normal University, Xi’an 710062, China; 2University of Chinese Academy of Sciences, Beijing 100049, China; 3Key Laboratory of Plant Resources Conservation and Sustainable Utilization, South China Botanical Garden, Chinese Academy of Sciences, Guangzhou 510650, China; 4National Maize Improvement Center, College of Agronomy and Biotechnology, China Agricultural University, Beijing 100094, China

**Keywords:** *Scutellaria baicalensis*, R2R3-MYB, expression profiles, abiotic stresses

## Abstract

R2R3-MYB transcription factors participate in multiple critical biological processes, particularly as relates to the regulation of secondary metabolites. The dried root of *Scutellaria baicalensis* Georgi is a traditional Chinese medicine and possesses various bioactive attributes including anti-inflammation, anti-HIV, and anti-COVID-19 properties due to its flavonoids. In the current study, a total of 95 *R2R3-MYB* genes were identified in *S. baicalensis* and classified into 34 subgroups, as supported by similar exon–intron structures and conserved motifs. Among them, 93 *R2R3-SbMYB**s* were mapped onto nine chromosomes. Collinear analysis revealed that segmental duplications were primarily responsible for driving the evolution and expansion of the *R2R3-SbMYB* gene family. Synteny analyses showed that the ortholog numbers of the *R2R3-MYB* genes between *S. baicalensis* and other dicotyledons had a higher proportion compared to that which is found from the monocotyledons. RNA-seq data indicated that the expression patterns of *R2R3-SbMYBs* in different tissues were different. Quantitative reverse transcriptase-PCR (qRT-PCR) analysis showed that 36 *R2R3-SbMYBs* from different subgroups exhibited specific expression profiles under various conditions, including hormone stimuli treatments (methyl jasmonate and abscisic acid) and abiotic stresses (drought and cold shock treatments). Further investigation revealed that *Sb*MYB18/32/46/60/70/74 localized in the nucleus, and *Sb*MYB18/32/60/70 possessed transcriptional activation activity, implying their potential roles in the regulatory mechanisms of various biological processes. This study provides a comprehensive understanding of the *R2R3-SbMYBs* gene family and lays the foundation for further investigation of their biological function.

## 1. Introduction

Transcription factors (TFs), as significant regulators at the transcriptional level, can control or affect diverse biological processes via activating or inhibiting specific gene expressions [1]. The myeloblastosis (MYB) TFs, as one of the largest functional TF families among plants, were characterized by an extremely conserved MYB DNA binding domain that is composed of one to four imperfect repeats at the N-terminus [2]. Each repeat is typically composed of 50–53 amino acids that encode three α-helices, the second and third of which create a 3D helix–turn–helix (HTH) structure via three regularly spaced tryptophan residues for binding to the major DNA groove at a specific recognition site during transcription [2]. There are four classes of MYB proteins that can be distinguished based on the number of Rs (MYB repeats), namely 1R-MYB, R2R3, 3R, and 4R-MYB [3]. Among them, the R2R3-MYB TFs that play important roles in a multitude of key physiological and biochemical processes, including organ development, hormone synthesis and signal transduction, response to biotic and abiotic stresses, and primary and secondary metabolism particularly those that affect nutritional and medicinal active ingredients or morphological and qualitative characteristics [4,5,6,7].

Recently, as many plants have undergone whole-genome sequencing, the number of identified *R2R3-MYB* gene members continues to increase. For example, 126 *R2R3-MYB* genes have been reported in *Arabidopsis thaliana* [8], 157 in *Zea mays* [9], 125 in *Oryza sativa* [10], 108 in *Vitis vinifera* [11], 110 in *Salvia miltiorrhiza* [12], 109 in *Hypericum perforatum* [13], 244 in *Glycine max* [14], 196 in *Populus trichocarpa* [15], and 69 in *Ginkgo biloba* [16]. Subsequently, the genetic and biochemical characterization of R2R3-MYB proteins in a wide variety of plant species has followed. Numerous R2R3-MYBs have been implicated in the response to biotic and abiotic conditions. For instance, rice *Os*MYB2 was involved in enhanced cold, salt, and dehydration tolerance and reduced ABA sensitivity [17]; *Mdo*MYB121 enhanced the tolerance to cold, drought, and salinity stress in *M. domestica* [18]; apple *Md*MYB88/124 is involved in cold tolerance by CBF-dependent and CBF-independent pathways [19]; and *AtMYB2*, *AtMYB102,* and *AtMYB41* were induced by salt, phosphate starvation, drought, and wounding stress in *Arabidopsis* [20,21,22].

Some R2R3-MYB TFs have been characterized in regulating secondary metabolites biosynthesis, especially anthocyanins and flavonoids in the phenylpropanoid biosynthetic pathway. For example, R2R3-MYB TFs in many species were identified and characterized for their participation in anthocyanin biosynthesis, including *AtMYB75/90/113*/*114* in *Arabidopsis* [23,24], *MdMYBA/1/3/10/110a* in apple [25,26,27], *PyMYB10/114* in pears [28,29], *FaMYB10* in strawberry [30], *StAN1* in potato [31], and *MaAN2* in *Muscari armeniacum* [32]. *AtMYB11*/*12*/*111* promotes flavonol accumulation in the *A. thaliana* seedling [24]. *EsMYB9* and *EsMYBF1* from *Epimedium sagittatum* positively regulate flavonoid biosynthesis and accumulation by activating the structural genes *CHS*, *CHI*, and *F3H* [33,34]. While *AtMYB4* [35] and *MdMYB28* [36] are repressors of flavonoids in *Arabidopsis* and apple, respectively. *VvMYB84* [37] from grape negatively regulates flavonoid biosynthesis in a similar manner to *GmMYB100* from soybean [38]. Currently, the functions of *R2R3-MYBs* have only been identified and characterized in *Arabidopsis*, agricultural, and vegetable plants, while little is known about them in Chinese herbal medicines and commercial crops.

*Scutellaria baicalensis* Georgi is an important medicinal plant in the Lamiaceae family and is extensively used in China, Japan, and Southeast Asian countries [39]. *Scutellariae Radix* (the dried root of *S. baicalensis*) has been applied as an ethnobotanical herbs for more than 2000 years, since being recorded in Chinese ancient medical books [40]. Pharmacological studies have shown that *Scutellariae Radix* possesses various bioactivities (e.g., antiviral, antibacterial, neuroprotectant, and cardiovascular protectant, etc.) [40]. The major medically active compounds in *Scutellariae Radix* are the flavonoids, such as baicalin, wogonoside, and their aglycones [41]. Encouragingly, baicalin and baicalein have been proven to repress the replication of COVID-19 virus via inhibiting SARS-CoV-2 3CLpro (3C-like protease) [42,43]. In this species, the genome sequence has been released and several enzyme genes of the flavonoid pathway have been identified [44,45], while the information of *R2R3-MYB* genes in *S. baicalensis* is lacking, which hinders research into *R2R3-SbMYBs* functionality to a certain extent. Hence, we performed a systematic bioinformatics analysis of *R2R3-SbMYB* gene family to understand the R2R3-MYB TF gene family′s function, evolution, and expression profiles in *S. baicalensis*.

Here, we carried out a genome-wide analysis for the *R2R3-MYB* gene family and discovered 95 members in *S. baicalensis*. The comprehensive bioinformatics analysis of the *R2R3-SbMYBs* was conducted, such as sequence features, phylogenetic relationships, chromosome distribution, gene structure, motif composition, cis-element analysis, and collinearity and synteny analysis. RNA-Seq analysis revealed the expression patterns of *R2R3-SbMYB* genes in different tissues. A total of 36 *R2R3-SbMYBs* were selected and subjected to gene expression analyses under hormonal treatments and abiotic stresses via qRT-PCR. Considering these results, we might be able to carry out a further functional study of R2R3-MYB proteins in *S. baicalensis* using these data.

## 2. Results

### 2.1. Identification, Sequence Feature, and Phylogenetic Analysis of the R2R3-SbMYB Members

A total of 209 *SbMYB* candidate genes containing complete MYB domains were originally identified from the *S. baicalensis* genome. Finally, 95 *R2R3-SbMYBs* genes were verified via Pfam and SMART. As a result of their random distribution, 93 *R2R3-SbMYB* genes have been renamed from *SbMYB1* to *SbMYB93* according to where they map on the chromosomes, while two *R2R3-SbMYBs* (evm.model.contig391.4 and evm.model.contig588.2), were renamed *SbMYB94* and *SbMYB95*, respectively, because they could not be assigned to any specific chromosome.

Multiple sequence alignment analysis revealed that R2R3-*Sb*MYB has a similar distribution of R2 and R3 repeat residues as *Arabidopsis* R2R3-MYB (Figure 1). The R2 repeat in both *S. baicalensis* and *Arabidopsis* contains three extremely conserved tryptophan residues (W) at positions 5, 25, and 45 (Figure 1A,B), which play significant roles in MYB-DNA interaction via forming a hydrophobic core in the HTH structure which is also known as the signature of the MYB domain. Similarly, in the R3 repeat, there are three highly conserved residues at positions 5, 24, and 43, including a phenylalanine (F) and two tryptophan residues (W) (Figure 1C,D), suggesting that the R2R3 domains were extremely conserved.

The detailed characteristics of the 95 R2R3-*Sb*MYB members were analyzed and are listed in Appendix A. The lengths of the amino acids of the R2R3-*Sb*MYB family ranged from 144 to 972, with the pI value ranged from 4.90 (*Sb*MYB62) to 9.85 (*Sb*MYB8) and MW from 10.82 kDa (*Sb*MYB58) to 91.76 kDa (*Sb*MYB57).

To further elucidate the functions of the R2R3-*Sb*MYB proteins and the evolutionary relationships within the R2R3-MYB family, an NJ tree containing 95 R2R3-*Sb*MYBs and 126 R2R3-*At*MYBs was constructed (Figure 2, Appendix A). These R2R3-MYBs from *S. baicalensis* and *A. thaliana* were divided into 34 subgroups (S1–S34) and named based on the previous report [8]. Remarkably, 29 clades contained different numbers of R2R3-MYBs from the two species, while four clades (S03, S10, S12, and S33) only included R2R3-*At*MYBs, and one clade (S29) was specific to R2R3-*Sb*MYBs (Figure 2), suggesting that these subgroup members may have particular responsibilities.

### 2.2. Analyses of Chromosome Distribution, Gene Duplication, and Synteny for R2R3-SbMYB Genes

According to genome chromosome location analyses, *R2R3-SbMYB* genes were unevenly spread throughout nine chromosomes (Figure 3). Chromosome 1 contained the largest number of *R2R3-SbMYBs* (28 members, accounting for 30.1%), followed by chromosome 2 with 15 *R2R3-SbMYB*s, while chromosome 7 contained the minimum number of three genes. Furthermore, these results suggested that the *R2R3-SbMYBs*, distributed adjacently on the same chromosome, were divided into the same subgroup in the phylogenetic analysis. For example, *SbMYB35*, *SbMYB36*, and *SbMYB37*, clustered and distributed on chromosome 2 (Figure 3), were grouped into the S9 (Figure 2), suggesting that these clustered genes may have similar molecular functions.

The collinearity of the *R2R3-SbMYB* genes was constructed via BLASTP and MCScanX software to understand the potential duplication events between them. Among these *R2R3-SbMYBs*, 25 segmental duplication pairs were detected between the *R2R3-SbMYBs* (Figure 4 and Appendix A). Meanwhile, we also observed intrachromosomal duplications, identified only one tandem duplication event with three *R2R3-SbMYBs*, in which *SbMYB35* was tandemly duplicated with *SbMYB36* and *SbMYB37* on two chromosome (Figure 3 and Appendix A). These results determined that gene duplications, especially segmental duplications, fueled *R2R3-SbMYBs* diversification and evolution.

To further elucidate the possible evolutionary relationship of the *R2R3-SbMYBs* between *S. baicalensis* and other eight representative species, a comparative syntenic map was constructed between these species (Figure 5 and Appendix A). A total of 70 *R2R3-SbMYBs* exhibited a syntenic relationship between *S. baicalensis* and *Sesamum indicum*, followed by *P. trichocarpa* (67), *V. vinifera* (54), *Solanum tuberosum* (44), *A. thaliana* (40), *Nicotiana attenuatattenuate* (28), *Z. mays* (15), and *O. sativa* (13). These results indicated that *S**. baicalensis* and dicots presented more homologous gene pairs than *S**. baicalensis* and monocots. Besides, some *R2R3-SbMYBs* were found to be collinear with at least three *R2R3-MYBs* of other species, especially those in *S. indicum* and *P. trichocarpa*, suggesting that these *R2R3-MYBs* may play important roles in the evolution of plants. Furthermore, these results showed that some *R2R3-SbMYBs*, including *SbMYB3*, *5*, *20*, *21*, *30*, *31*, *34*, *38*, and *50*, had collinear paired genes with *R2R3-MYBs* of other seven species, indicating that these synonymous gene pairs are highly conserved in *R2R3-MYB* evolution.

### 2.3. Gene Structure, Motif Composition, and Cis-Elements Analysis

Our study also constructed an unrooted phylogenetic tree based on 95 amino acid sequences of R2R3-SbMYBs in order to better understand their gene structure, conserved motifs, and cis-elements (Figure 6A). Gene structure analysis revealed that different intron numbers have been found in *R2R3-SbMYBs*, ranging from 0 to 15. Approximately 86% of genes contain one to three introns, with the majority containing two introns. Some *R2R3-SbMYBs* had a large number of introns, with 15, 11, and 10 introns in *SbMYB94*, *SbMYB39*, and *SbMYB14*, respectively (Figure 6B). In addition, we also found that similar exon/intron structures were observed among R2R3-*Sb*MYBs in the same subgroup.

We identified 20 conserved motifs in the R2R3-*Sb*MYB proteins (Figure 6C), which we referred to as motif 1 to 20. Interestingly, each R2R3-*Sb*MYB protein contained motif 1, 2, 3, 4, and 5, which were commonly conserved in *S. baicalensis*. Additionally, we found similar motif compositions among R2R3-*Sb*MYBs in the same subgroup, suggesting similar functions.

Cis-elements analysis demonstrated that a total of 73 kinds of elements were found in the promoter regions of 95 *R2R3-SbMYBs*, and the majority of which were related to hormone response and abiotic/biotic stress (Figure 6D and Appendix A). The abscisic acid (ABA) response element was the most common and existed in the promoter regions of 69 genes, followed by salicylic acid (SA) (58), methyl jasmonate (MeJA) (57), MYB binding sites (43), auxin (31), and defense and stress response elements (28), as showed Figure 6D and Appendix A. The above results indicated that *R2R3-SbMYBs* can be regulated by numerous factors and are possibly involved in hormone-regulation, development, and stress responses.

### 2.4. RNA-Seq Analysis of R2R3-SbMYB Expression in Various Tissues

To explore the tissue-specific expression profiles of *R2R3-SbMYBs*, we analyzed the transcriptional abundance of *R2R3-SbMYBs* in four tissues (root, stem, leaf, and flower) based on the RNA-seq data (unpublished) (Figure 7). The results indicated that 18 highly expressed *R2R3-SbMYBs* (FPKM > 5) were detected in all four tissues, of which *SbMYB74* and *SbMYB12* showed higher expression levels (FPKM > 50). In contrast, 27 *R2R3-SbMYBs* were expressed at low level (FPKM < 1) in the four different tissues, of which, the transcriptional levels of eight genes were extremely low, including *SbMYB9*/*37*/*41*/*43*/*59*/*67*/*87* (Appendix A). Furthermore, some *R2R3-SbMYBs* exhibited tissue-specific or preferential expression. For example, *SbMYB88*/*54*/*22* was preferably expressed in the stem, while *SbMYB3*/*64*/*75*/*81* were highly expressed in the flower (Figure 7). Interestingly, no *R2R3-MYB* was specifically expressed in the roots and leaves of *S. baicalensis* (Appendix A).

### 2.5. Expression Patterns of R2R3-SbMYBs under Hormone Stimuli and Abiotic Stresses

To explore whether the expression levels of *R2R3-SbMYBs* were affected by hormone treatments or abiotic stresses, we selected 36 members from different subgroups to determine their expression levels by qRT-PCR based on the analysis results of known R2R3-MYB proteins and cis-elements under hormone stimuli treatments (ABA and MeJA) and abiotic stresses (4 °C and PEG6000). In summary, the results showed that *SbMYB44*/*52*/*78* expression levels were affected in all four stress conditions, while *SbMYB63* and *SbMYB55* were not obvious in all treatments.

For hormone stimuli, *SbMYB12*/*44*/*52*/*64/74*/*75*/*78* showed similar expression patterns, all of which were significantly increased under ABA and MeJA treatments (Figure 8 and Figure 9). Specifically, the expression levels of *SbMYB30*/*44*/*47*/*64*/*71* also increased significantly under ABA stress (Figure 8), while the transcription levels of *SbMYB21*/*29*/*49*/*58*/*60/62* also increased significantly under MeJA treatment (Figure 9). In addition, the expression levels of *SbMYB5*/*14*/*22*/*49*/*53*/*70*/*85* decreased significantly under ABA stimuli (Figure 8), and *SbMYB5*/*14*/*45*/*66*/*73*/*89*/*92* decreased significantly under MeJA treatment (Figure 9). This result indicates that ABA and MeJA treatments had significant effects on the expression of these genes.

For abiotic stresses, the expression levels of *SbMYB44*/*52*/*78* were significantly increased under both drought (PEG6000) and low temperature (4 °C) treatments (Figure 10 and Figure 11). Particularly, the expression levels of *SbMYB12*/*32*/*47*/*49*/*70*/*91* increased significantly under PEG6000 stress (Figure 10), and *SbMYB5*/*26*/*58*/*66*/*71*/*89* increased significantly under low temperature stress (Figure 11), while the expression levels of *SbMYB29*/*45*/*58*/*82* decreased significantly under drought conditions (Figure 10), and *SbMYB29*/*46*/*55*/*62*/*64*/*73*/*85*/*91*/*92*/*93* decreased significantly under low temperature stress (Figure 11).

### 2.6. Subcellular Localization and Trans-Activating Assays of R2R3-SbMYBs

To further elucidate the potential function of *R2R3-SbMYBs* in the transcriptional regulation system, we amplified the coding regions of the selected genes, *SbMYB18*/*32*/*46*/*60*/*70*/*74* and fused them to HBT-GFP-NOS vector, respectively. The primers for vector construction are shown in Appendix A. As shown in Figure 12, GFP protein was expressed in the nucleus and cytoplasm as a control, while the *Sb*MYBs-GFP were specifically localized within the nucleus, which was consistent with our prediction (Appendix A). Similar to many other TFs, *Sb*MYB18/32/46/60/70/74 may be involved in the transcriptional regulation system.

As part of our investigation into whether these proteins have transactivation activity, we fused *Sb*MYBs in pGBKT7 to produce the recombinant vector pGBKT7-*Sb*MYBs and conducted transactivation assays in the yeast strain AH109. As shown in Figure 13, all the yeast cells containing pGBKT7-*Sb*MYBs or control plasmid pGBKT7 grew well on the SD/-Trp medium. However, only *Sb*MYB18/32/60/70 grew normally and turned blue on the SD/-Trp/-His/-Ade medium with X-α-gal. These results indicate that *Sb*MYB18/32/60/70 exhibited transcriptional activation in yeast, and that these proteins were functional transcription factors.

## 3. Discussion

*Scutellaria baicalensis*, as a Chinese medicinal material, was traditionally used to treat lung and liver diseases in many Asian countries [40]. In recent years, *Scutellariae Radix* has been becoming one of the top ten sales of medicinal herbs in the Chinese medicine market because of its growing market demand every year and huge medicinal value [46]. R2R3-MYB proteins are involved in various plant biological processes including organ development processes, hormone synthesis and signal transduction, response to biotic and abiotic stresses, and primary and secondary metabolism, which were the main forms in all higher plants [4,5,6,7]. As a result of the release of the whole genomes of numerous plant species, a genome-wide identification of R2R3-MYB genes was performed in an increasing number of them [4,13,14,16,18,47,48,49]. As of yet, no research has been conducted on the systematic identification and investigation of the *R2R3-MYB* gene family in *S. baicalensis.*

Our study identified 95 genes that were associated with the *R2R3-MYB* family in *S. baicalensis*. The amount of *R2R3-SbMYB* was higher than the identified *R2R3-MYB* genes in ginkgo (69), and cucumber (71), it was less than that of *Arabidopsis* (126), rice (125), Salvia (110), and potato (111), especially significantly less than that of poplar (192), and soybean (244). This suggested that these species evolved with gene duplication and functional differentiation [13]. Multiple sequence alignment showed that all the 95 R2R3-*Sb*MYB proteins possessed a highly conserved MYB domain (Figure 1), while the C-terminal region of the MYB domain contained plentiful variations.

Based on the phylogenetic tree, 34 subgroups (S1–34) were identified between *S. baicalensis* and *Arabidopsis R2R3-MYB* genes (Figure 2). As a result of their similar amino acid sequences, proteins from the same subgroup usually perform similar functions in plant physiological and biochemical processes [9]. Therefore, we speculated that *Sb*MYB18 and *Sb*MYB32 had the similar function as *At*MYB75 and *At*MYB90 (S6), which were known for their role in promoting anthocyanins accumulation [23,24]. *At*MYB11/12/111 (S7) is involved in regulating the biosynthesis of flavanols [8], which hinted that *Sb*MYB46 and *Sb*MYB70 in S7 may regulate flavanol biosynthesis. *Sb*MYB12 may regulate anthocyanin and flavonoid accumulation, similar to *At*MYB112 in S20 [50]. Additionally, we also found that *Sb*MYB58 protein clustered in subgroup S29 alone, suggesting that it should be conferred the particular functions which was either lost in *Arabidopsis* or acquired in *S. baicalensis* after divergence from the last common ancestor. Similar phenomena were observed in previous studies [13,15]. In addition, some subgroups (S03, S10, S12, and S33) only incorporate R2R3-*At*MYBs from *Arabidopsis*, which suggested that these genes may be lost in the evolution of *S. baicalensis*. As an example, no *SbMYBs* were classified into *Arabidopsis* S12, the members of which regulate glucosinolate synthesis. Glucosinolates mainly exist in Brassicaceae plants and were secondary metabolites containing nitrogen and sulfur that can act as insecticides and prevent herbivore effects by causing bitterness and pungency. Similar results were found in *Beta vulgaris* [51], *P. bretschneideri* [52], and *P. trichocarpa* [15], just as in *S. baicalensis*. These insights will facilitate further studies on the functions of *R2R3-SbMYBs*.

Gene duplication is a major factor that contributes significantly to evolutionary expansion and genetic diversity in the plant kingdom [53]. Previous studies found that segmental duplications were the main mechanism by which the *R2R3-MYB* gene family expanded in apple, potato, and *Pistacia chinensis* [48]. In our study, 25 segmental duplication pairs were found (Figure 4 and Appendix A), and only one tandem duplication event was detected. These results indicated that gene duplications, especially segmental duplications, may contribute to the expansion of *R2R3-SbMYB* genes [54]. Here, we also observed that all the three tandemly duplicated genes (*SbMYB35*, *SbMYB36*, and *SbMYB37*) had two introns and were grouped in the same subfamily (S9) with *AtMYB16* (Figure 6B). It suggests that these genes may have similar functions to *AtMYB16* in controlling cell and petal morphogenesis in *S. baicalensis* [55]. Synteny analyses showed that the number of orthologs between *S. baicalensis* and those dicotyledons (*A. thaliana*, *P. trichocarpa*, *V. vinifera*, *S. tuberosum*, *N. attenuate,* and *S. indicum*) was more comparable to those between *S baicalensis* and the monocotyledons (*O. sativa* and *Z. mays*) (Figure 5 and Appendix A), suggesting that these orthologous pairs diverged during the evolution of different plants [16]. Furthermore, we also found that some synonymous gene pairs of *R2R3-SbMYBs* are highly conserved in *R2R3-MYB* evolution, which is in accordance with previous studies [16].

Gene structure analysis indicated that most *R2R3-SbMYB* genes had one to three introns (86%) and the majority of which contained two introns (Figure 6B). Notably, *SbMYB39* and *SbMYB94* in the S26 subgroup have complex structures which contained 11 and 15 introns (Figure 6B), respectively, similar to their orthologs gene *AtMYB124* and *AtMYB88* in *Arabidopsis* [56]. Moreover, MEME analysis indicated that the majority of R2R3-*Sb*MYBs in the same subgroup had conserved motif composition outside the MYB domains, suggesting that they have similar functions (Figure 6C). These results indicate that *R2R3-SbMYB* gene evolution occurred in a conservative pattern, rather than through accidental mutations.

To understand the function of these genes, it is crucial to examine the abundance of *R2R3-SbMYB* transcripts in different tissues. Based on RNA-seq data, we investigated the expression patterns of *R2R3-SbMYB* genes in four tissues. Some tissue-specific *R2R3-SbMYBs* as well as highly-expressed members were identified (Appendix A), which may play important roles during the growth and development of *S. baicalensis*. The results indicated that 18 highly expressed *R2R3-SbMYBs* were detected in all four tissues, of which *SbMYB74* and *SbMYB12* showed higher expression levels (FPKM > 50). In addition, further studies showed that they had a close phylogenetic relationship with *AtMYB73* (S22) and *AtMYB112* (S20), respectively, indicating that they may have similar functions [8]. *SbMYB18* showed the preferential expression in roots and was phylogenetically closer to *AtMYB75* and *AtMYB90*, suggesting that this gene possibly regulated the production of anthocyanins in root tissues. It was noteworthy that two *R2R3-MYB* genes (*SbMYB3* and *SbMYB81*) were expressed specifically in flowers and phylogenetically sub-grouped in S19 with *AtMYB21*/*22*, suggesting *Sb*MYB3 and *Sb*MYB81, as flower-specific transcription factors, may have a similar functional characteristics in participating in the regulation of the *FLS1* gene in anther and pollen [57,58].

Several cis-elements were identified in the promoters of *R2R3-SbMYB* genes, such as ABA, SA, MeJA, light, cold, and drought responsive elements, suggesting that these genes are involved in responses to various hormones and environmental stresses (Figure 6D and Appendix A). As is well known, plants sense and respond to various stress conditions by producing numerous secondary metabolites, which is an important mechanism for plants to survive under adverse conditions. It has been shown that MeJA and ABA are effective elicitors and can facilitate the accumulation of medicinal plant active ingredients [59,60]. Appropriate drought stress can significantly promote the accumulation of baicalin by stimulating the expression of the key enzymes in *S. baicalensis* [61]. Low temperature stress has a similar promoting effect on the accumulation of flavonoids [62]. Therefore, we determined the expression levels of 36 *R2R3-SbMYB**s* under MeJA, ABA, drought, and low temperature conditions via qRT-PCR. As shown in Figure 8, Figure 9, Figure 10, and Figure 11, some *R2R3-SbMYB**s*, especially *SbMYB44*/*52*/*78*, can be induced synchronously under various stress conditions, indicating that they are pleiotropic regulators which mediate the cross-talk between multiple signaling pathways under hormone stimuli treatments and abiotic stresses. In our present work, *SbMYB74* responded to ABA, MeJA, and cold treatment and phylogenetically clustered with ABA-mediated stress-related subgroup S22 (*AtMYB70*/*73*/*77*/*44*) of *Arabidopsis* [8], suggesting that it may have potential value in the stress responses of *S. baicalensis.* Of note, the transcription level of *SbMYB12* was significantly activated under MeJA, ABA, and drought treatments, hinting that it may play important roles in the growth and development of *S. baicalensis*, particularly in the regulation of the biosynthesis of secondary metabolites. Additionally, a significant number of R2R3-SbMYB genes play a role in response to only one stress stimulus, indicating that there may be distinct signaling pathways that are involved in the response to stress. Taking *SbMYB60* as an example, the gene only responded to MeJA treatment and was preferentially expressed in the stem tissues, which had a close phylogenetic relationship with *AtMYB0*/*23*/*66* (S15), suggesting that it possibly involved in determining epidermal stem cell types [8].

Systematic genomic bioinformatics analysis based on the whole genome data, combined with transcriptome analysis in various tissues and qRT-PCR results under abiotic stresses or hormone responses, affords valuable information for identifying valuable candidate genes for further functional characterization. Additionally, we performed subcellular localization and transactivation analysis for six selected genes, *Sb*MYB18/32/46/60/70/74. The results showed that these six proteins were localized in the nuclei (Figure 12), among which *Sb*MYB18/32/60/70 exhibited transcriptional activation activity in yeast (Figure 13).

## 4. Materials and Methods

### 4.1. Identification of R2R3-MYB Transcriptional Factors in S. baicalensis 

From the Pfam database (http://pfam.xfam.org/family/PF00249, accessed on 12 August 2022), we downloaded the Hidden Markov Model (HMM) profile of the MYB DNA binding domain (PF00249), which was used to search *MYB* genes against the genome of *S. baicalensis*. Candidate MYB proteins were confirmed using the Pfam and SMART database (http://smart.embl.de/, accessed on 12 August 2022) [13] and only those with complete R2R3 domains were identified as members of the R2R3-MYB family. An analysis of the physicochemical properties of the R2R3-SbMYB proteins was conducted with the ExPASy server (http://web.expasy.org/-compute_pi/, accessed on 12 August 2022) [53], which included measuring their sequence lengths, theoretical isoelectric points (pI), and molecular weight (MW). Sequence logos for R2 and R3 MYB domains in 95 R2R3-SbMYB proteins were created by using the WEBLOGO online program (http://weblog.berkeley.edu/logo.cgi, accessed on 12 August 2022).

### 4.2. Chromosome Distribution, Gene Duplication, Synteny, and Phylogenetic Analyses of R2R3-SbMYB Genes

Using the *S. baicalensis* genome database, chromosome distribution data for *R2R3-SbMYB* genes was retrieved, and the chromosome position map was generated by Tbtools software [63]. We examined gene duplication events using the Multiple Collinear Scanning Toolkit (MCScanX) [64], which was visualized by Circos software [65]. Syntenic analysis maps were generated using Dual Synteny Plotter software (https://github.com/CJ-Chen/TBtools, accessed on 12 August 2022) [63] for investigating the synteny relationships among R2R3-MYBs in *S. baicalensis* and other species (*A. thaliana*, *O. sativa*, *Z. mays*, *S**. tuberosum*, *P. trichocarpa*, *V. vinifera*, *N**. attenuata*, and *S**. indicum*).

Additionally, the protein sequences of 126 R2R3-MYBs from *A. thaliana* were downloaded from the TAIR (http://www.arabidopsis.org/, accessed on 12 August 2022). The protein sequence of 95 R2R3-*Sb*MYBs and 126 R2R3-*At*MYB were used for phylogenetic analysis. With the MEGA version 7.0 [66], a neighbor-joining (NJ) phylogenetic tree was constructed and the multiple sequence alignments were performed with ClustalW [67]. Based on the topology of the phylogenetic tree, the R2R3-SbMYB proteins were separated into several groups.

### 4.3. Gene Structure, Motif Composition, and Cis-Elements Analysis

The exon-intron structures of the *R2R3-SbMYB* genes were graphically displayed via TBtools software according to the CDS and genome sequence [63]. We predicted the conservation motifs by using the MEME program [68], where 20 motifs were the maximum number and 6 to 100 was the optimum width. To identify the cis-elements in the *R2R3-SbMYB* promoter regions, the 2000 bp genomic DNA sequences that were upstream of the start codon (ATG) were submitted to the PlantCARE database (http://bioinformatics.psb.ugent.be/web-tools/plantcare/html/, accessed on 12 August 2022).

### 4.4. Plant Materials and Various Treatments

The seeds of *S. baicalensis* were collected from Longxi County, Gansu Province, China. The seedlings were cultivated in the incubator at 25 ± 2 °C (16 h light/8 h dark cycle, 65 ± 5% humidity). Seedlings that were two-months-old in a uniform growth state were subjected to abiotic stresses and hormonal treatments. For hormonal treatments, young seedlings were sprayed with 200 µM MeJA or 100 µM ABA as described previously [69]. For drought stress, the seedlings were placed in pots with 20% PEG6000. For cold shock treatment, the seedlings were exposed to 4 °C. The seedlings were treated for 0 (as the control); 0.5, 1, 2, 3, and 6 h under MeJA and ABA treatments; and 0 (as the control), 1, 3, 6, 12, and 24 h under PEG6000 and cold treatments. After collecting the samples, they were immediately frozen in liquid nitrogen and stored at −80 °C until RNA extraction was performed. A total of three biological replicates were collected. All primers that were used in the study are listed in Appendix A.

### 4.5. RNA-Seq Data and qRT-PCR Analysis

During the flowering stage of *S. baicalensis*, the roots, stems, leaves, and flowers were collected to sequence its transcriptome. The expression profiles of *R2R3-SbMYB* were hierarchically clustered with average linkage based on the FPKM value, and displayed by the Heatmap illustrator of TBtools [63].

The procedure for total RNA isolation and qRT-PCR analysis has been described previously [70]. The *SbACT7* (evm.TU.contig159.4) gene was used as the reference gene [70]. Based on the qRT-PCR data, the expression level of the corresponding gene was calculated using the 2^−ΔΔCt^ method.

### 4.6. Subcellular Location Analyses of SbMYBs

To study the subcellular location of *Sb*MYBs protein, the coding sequence of *SbMYBs* (including *SbMYB18*/*32*/*46*/*60*/*70*/*74*) lacking the stop codon were amplified and fused with the HBT-GFP-NOS vector to construct the *Sb*MYBs-GFP fusion protein expression vector. Mesophyll protoplasts were prepared from the leaves of *Arabidopsis* that were grown under 12 h light/12 h darkness for four weeks and transformed as previously reported [69]. We first incubated the transformed protoplasts at 21 °C for 12 h, and then observed them under a confocal laser microscope with high resolution (Leica TCS SP5, Wetzlar, Germany).

### 4.7. Transcriptional Activation Assays of SbMYBs

The ORF of *SbMYBs* (including *SbMYB18*/*32*/*46*/*60*/*70*/*74*) was cloned and integrated into the pGBKT7 vector using the Gateway recombinational cloning system [71]. The recombinant vector BD-*SbMYBs* was transformed into the strain AH109 and cultured on SD/-Trp at 28 °C for 2–3 days, then screened on SD/-Trp/-Ade/-His/X-α-gal media to assay the transactivation activity.

## 5. Conclusions

Here, 95 *R2R3-MYB* gene family members were identified in the genome of *S. baicalensi*, followed by a comprehensive analysis, including chromosome distribution, gene duplication and synteny, gene structure, motif composition, and cis-elements analysis. Phylogenetic comparisons, transcriptome analysis in various tissues, and qRT-PCR results under various conditions predicted the potential functions of these *R2R3-SbMYB* genes. We also analyzed the subcellular localization and transactivation of six key model genes. Together, these results provided novel insights into the functionality of *R2R3-SbMYB* in flavonoid biosynthesis and established a foundation for further investigation. Additionally, the assumptions about the functionality of the R2R3-MYB gene in this study provide directions for future related proof research.

## Figures and Tables

**Figure 1 ijms-23-09342-f001:**
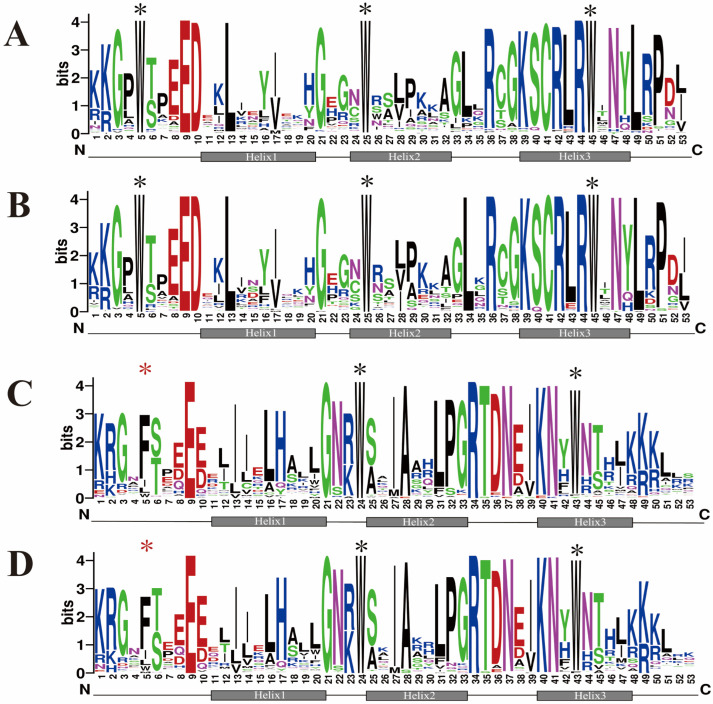
Comparison of the MYB R2 and R3 sequences in *S. baicalensis* and *Arabidopsis* R2R3-MYBs. (**A**) R2 sequence logos in *S. baicalensis* MYBs. (**B**) R2 sequence logos in *Arabidopsis* MYBs. (**C**) R3 sequence logos in *S. baicalensis* MYBs. (**D**) R3 sequence logos in *Arabidopsis* MYBs. Highly conserved tryptophan (W) and phenylalanine (F) residues are indicated by asterisks. To provide clues for the functional investigation of R2R3-*Sb*MYB proteins, we predicted subcellular localization and found 94 R2R3-*Sb*MYB proteins in the nuclei, while only *Sb*MYB66 was within the mitochondria.

**Figure 2 ijms-23-09342-f002:**
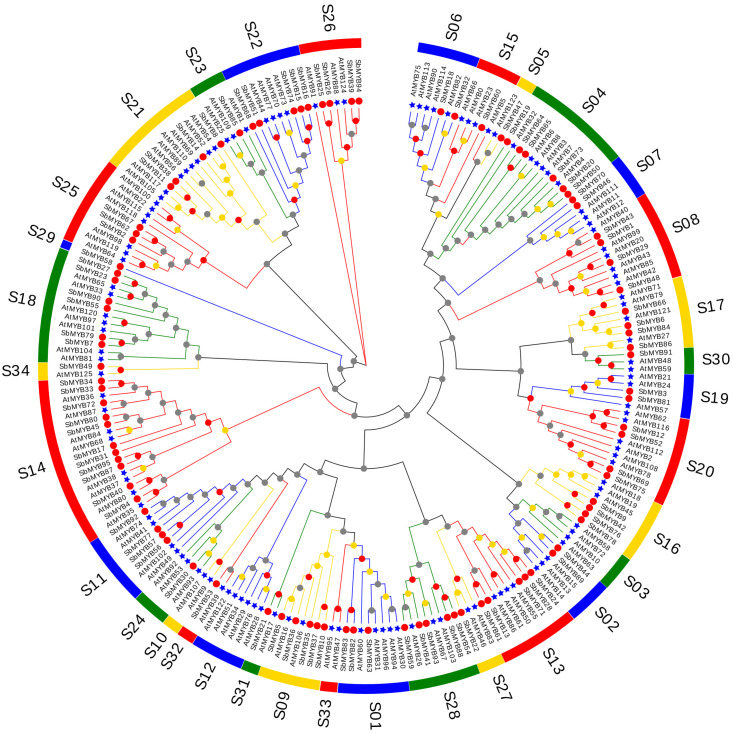
*S. baicalensis* (red) and *Arabidopsis* (blue) R2R3-MYB proteins phylogenetic relationships. A neighbor-joining (NJ) phylogenetic tree was generated using 1000 replications. A total of 34 subgroups of R2R3-MYBs (S1–S34) were determined along with *Arabidopsis* homologs. The red circle refers to SbMYBs and the blue star refers to AtMYBs, and the outer circle of the phylogenetic tree contained the ID number of each subgroup.

**Figure 3 ijms-23-09342-f003:**
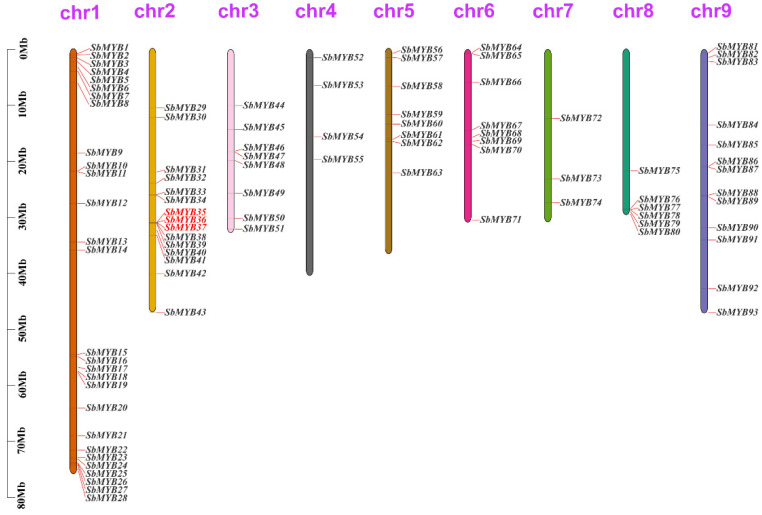
Chromosomal map of *R2R3-SbMYB* genes was generated for *S. baicalensis*. A total of 93 *R2R3-SbMYB* genes were located on nine different chromosomes, with the remaining two genes resided on the unassembled scaffold (*SbMYB94* and *SbMYB95*). Every chromosome has a number at the top and is scaled in megabases (Mb). Gene names that are highlighted in red on each chromosome indicate tandem duplications.

**Figure 4 ijms-23-09342-f004:**
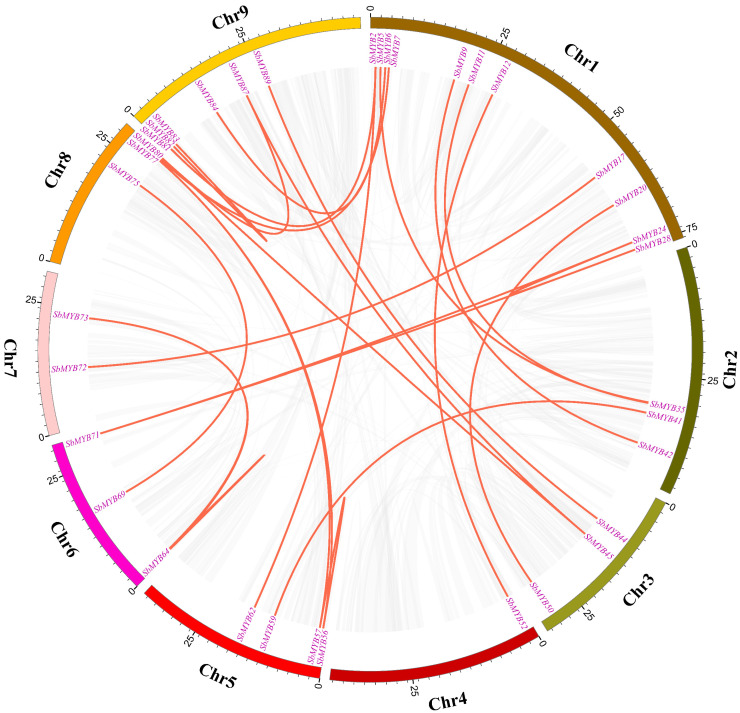
Interchromosomal relationships of the *R2R3-MYB* genes in *S. baicalensis*. Synteny blocks in *S. baicalensis* are depicted in gray lines, and duplicate MYB gene pairs are shown in red lines.

**Figure 5 ijms-23-09342-f005:**
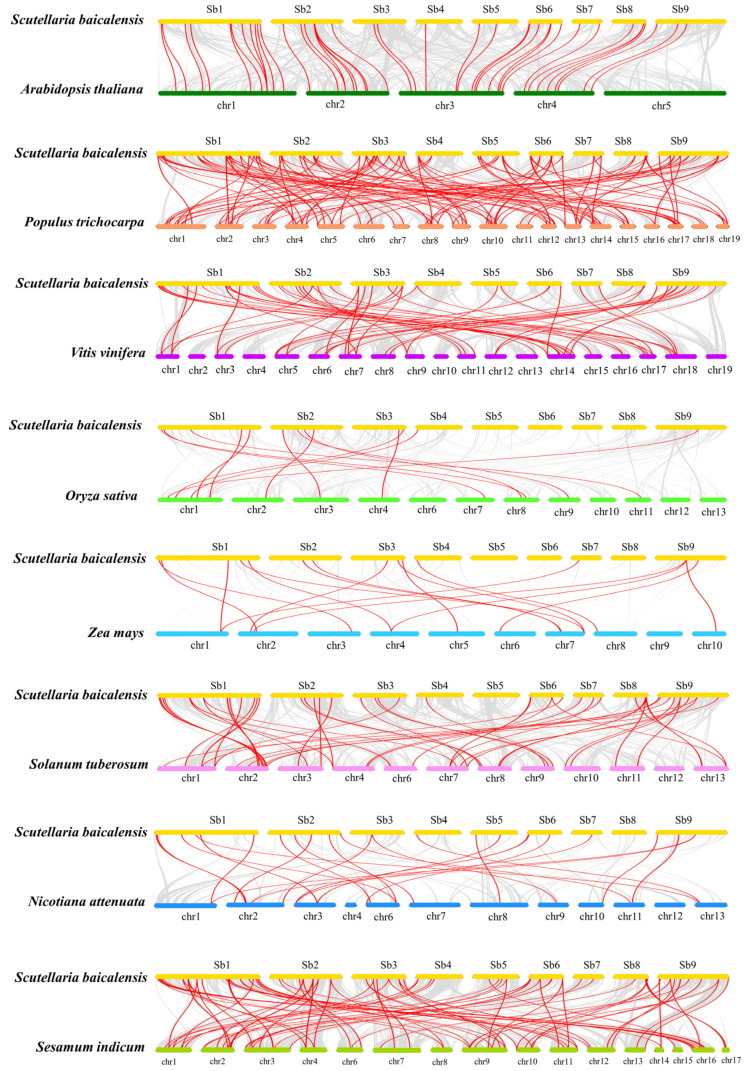
Synteny analysis of *S. baicalensis R2R3-MYB* genes in comparison with eight other representative plant species (*A. thaliana*, *P. trichocarpa*, *V.vinifera*, *S. tuberosum*, *N. attenuata*, *S. indicum O. sativa*, and *Z. mays*). Red lines highlight the syntenic *R2R3-MYB* gene pairs between *S. baicalensis* and the other plants, while gray lines illustrate all collinear blocks.

**Figure 6 ijms-23-09342-f006:**
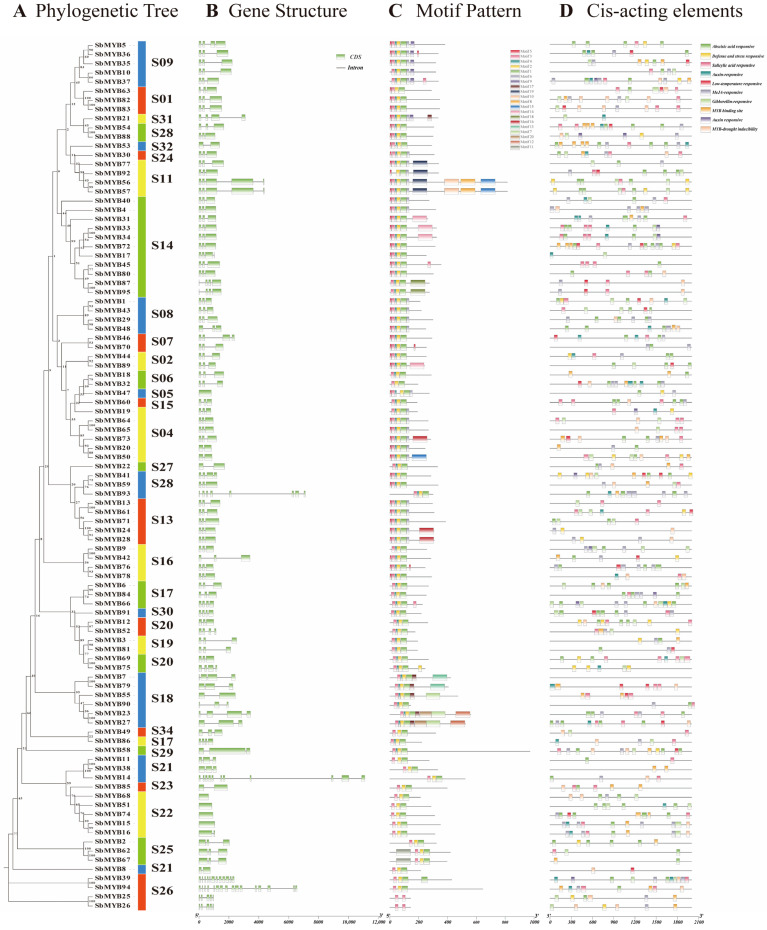
Phylogenetic analysis of the gene structures, conserved protein motifs, and cis-acting elements of R2R3-MYBs from *S. baicalensis*. (**A**) Phylogenetic tree of the R2R3-*Sb*MYB proteins. (**B**) Structures of *R2R3-SbMYB* genes are depicted with green boxes for exons and gray lines for introns. (**C**) A total of 20 motif compositions of the R2R3-*Sb*MYB proteins are represented via differently colored boxes. (**D**) Analysis of *R2R3-SbMYB* gene promoters for cis-acting elements.

**Figure 7 ijms-23-09342-f007:**
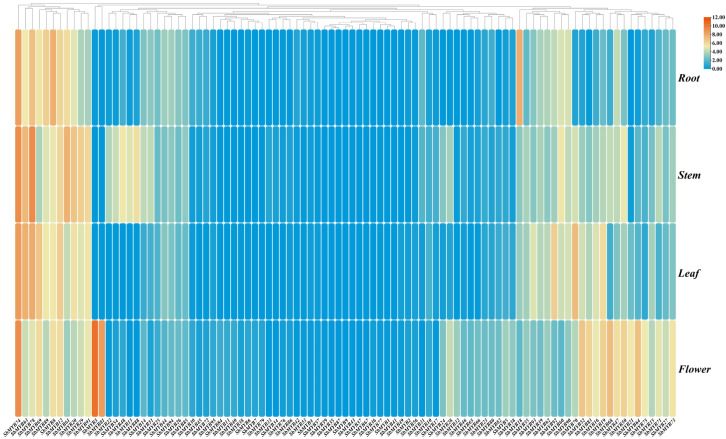
The heatmap shows the hierarchical clustering of expression profiles of *R2R3-SbMYB* genes in different tissues based on Lg (FPKM) values.

**Figure 8 ijms-23-09342-f008:**
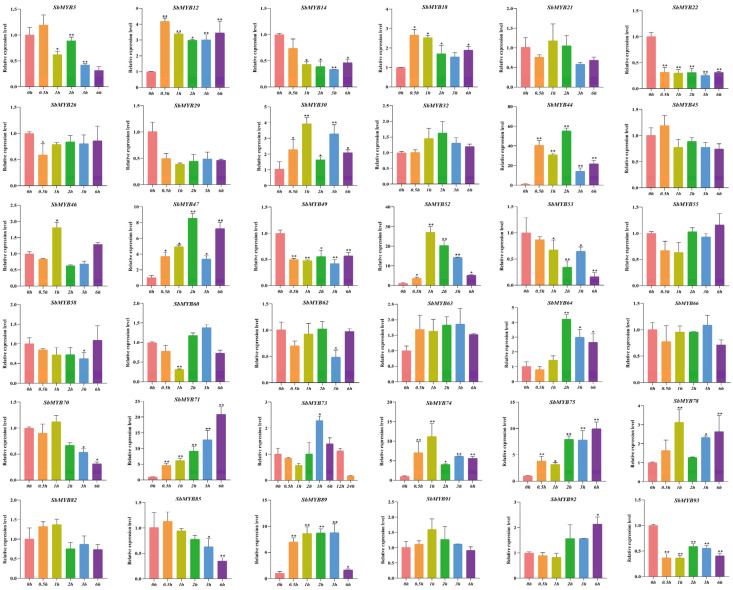
Response to ABA treatments of 36 selected *R2R3-SbMYB* genes. The reference gene was *SbACT7.* Comparing the experimental and control groups (0 h), statistically significant differences are indicated by * *p* < 0.05, ** *p* < 0.01.

**Figure 9 ijms-23-09342-f009:**
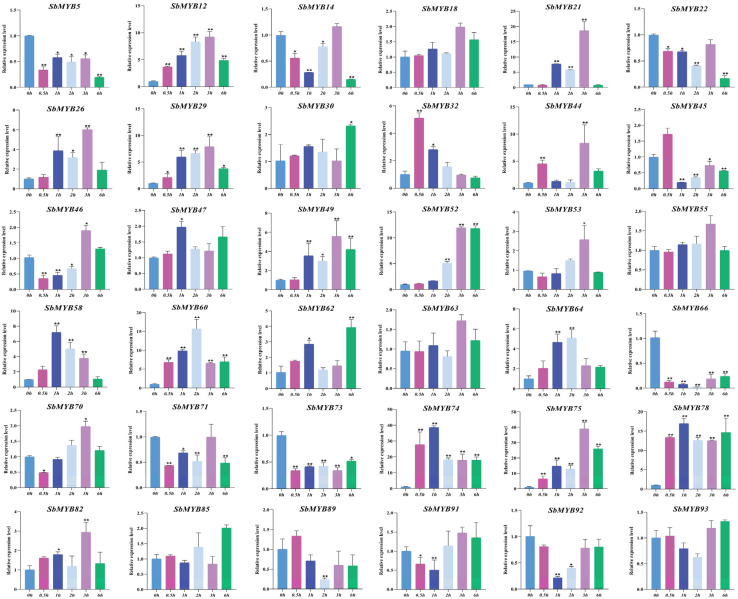
Response to MeJA treatments of 36 selected *R2R3-SbMYB* genes. The reference gene was *SbACT7.* Comparing the experimental and control groups (0 h), statistically significant differences are indicated by * *p* < 0.05, ** *p* < 0.01.

**Figure 10 ijms-23-09342-f010:**
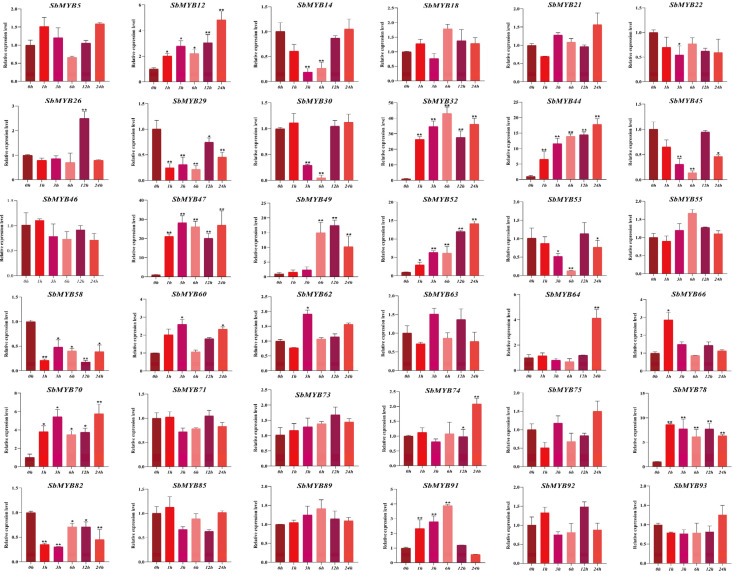
Response to PEG6000 treatments of 36 selected *R2R3-SbMYB* genes. The reference gene was *SbACT7.* Comparing the experimental and control groups (0 h), statistically significant differences are indicated by * *p* < 0.05, ** *p* < 0.01.

**Figure 11 ijms-23-09342-f011:**
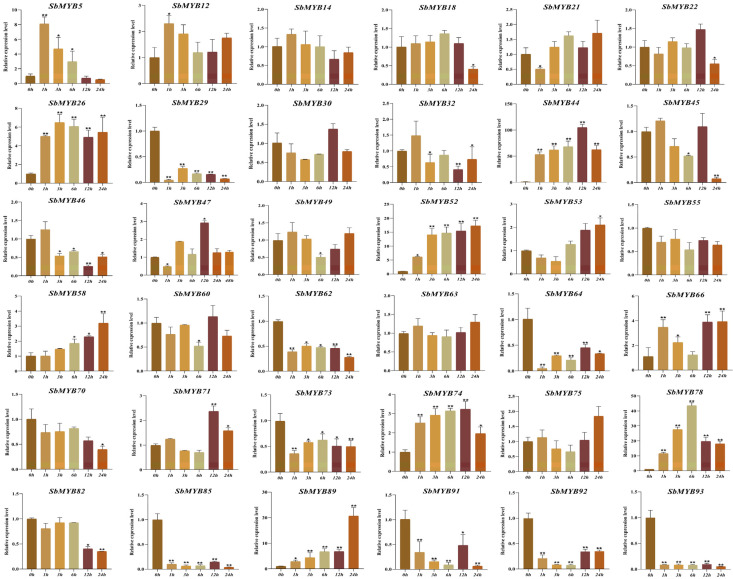
Response to low temperature treatments of 36 selected *R2R3-SbMYB* genes. The reference gene was *SbACT7.* Comparing the experimental and control groups (0 h), statistically significant differences are indicated by * *p* < 0.05, ** *p* < 0.01.

**Figure 12 ijms-23-09342-f012:**
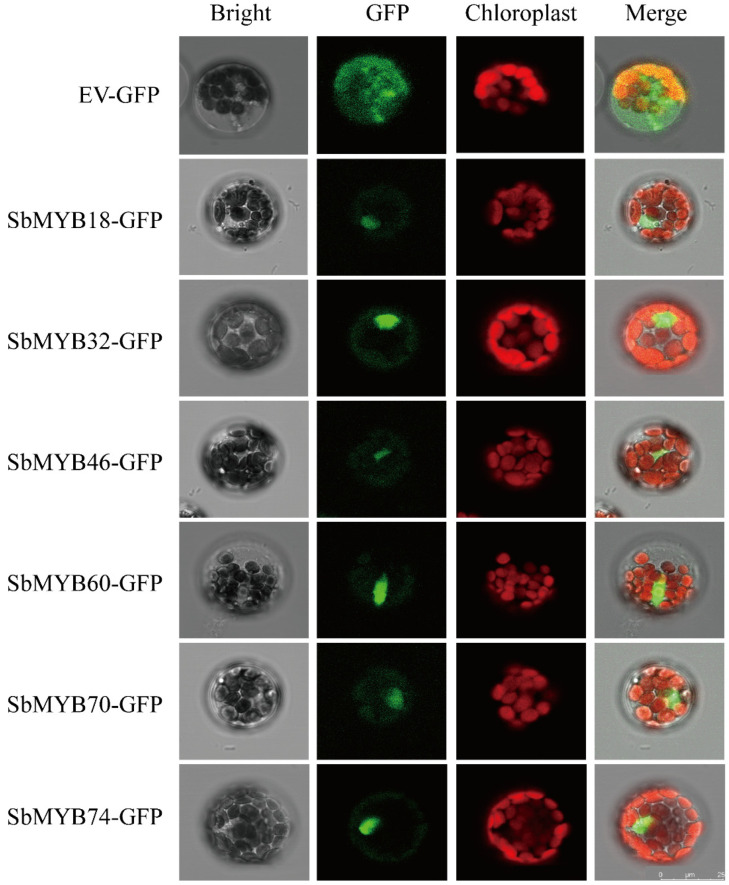
Subcellular location of the selected *Sb*MYBs (*Sb*MYB18/32/46/60/70/74) in *Arabidopsis* mesophyll protoplasts by laser confocal microscopy, with GFP as the positive control (Scale bar: 25 µm).

**Figure 13 ijms-23-09342-f013:**
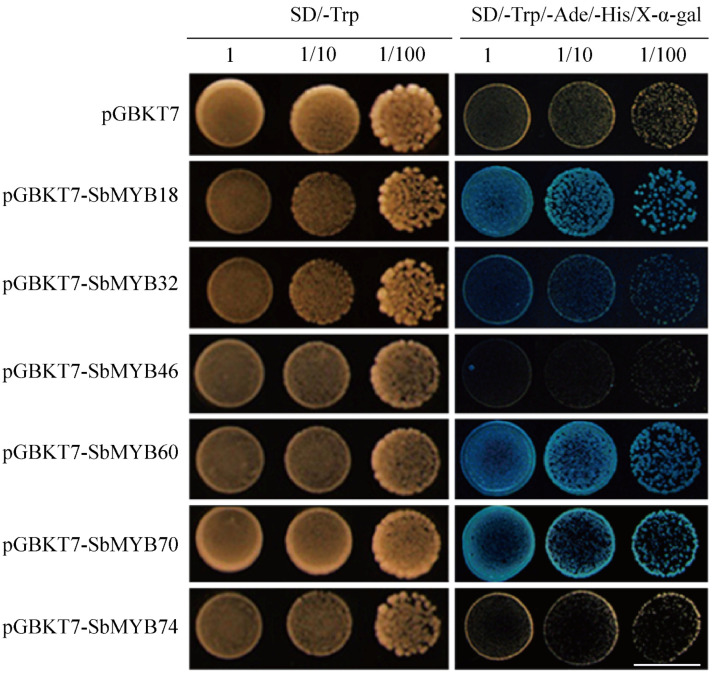
Transactivation activity of *Sb*MYBs protein. The construct of the six recombinant vector pGBKT7-*Sb*MYBs were transformed into AH109 and detected on SD/-Trp, SD/-Trp/-His/-Ade medium with X-α-gal. (Scale bar: 8 mm).

## Data Availability

MDPI Research Data Policies.

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
