# Peer review of "Systematic Analysis and Functional Characterization of R2R3-MYB Genes in Scutellaria baicalensis Georgi"

_ijms, 2022, doi:10.3390/ijms23169342_

Round 1

Reviewer 1 Report

Dear authors.

Many plants contain metabolites with biological activities. The development of bioinformatics methods has opened up prospects for determining the mechanisms of formation of the properties and characteristics of metabolites. The scientific content of the manuscript justifies its publication, but some additions and modifications will significantly improve the quality of the article.

Major comments:

1) In Introduction, the goal should be formulated.

2) L. 307 and in the text, at the beginning of the sentence the full biological name should be indicated without an abbreviation.

3) L.337, spacing should be.

4) All author assumptions about the functionality of the R2R3-Bmp gene require proof. This should be discussed in conclusion.

5) In the References, 47% of publications refer to 2017-2021 (the last 5 years); the remaining 53% of used sources are older than 5 years. It is recommended to increase the share of references to sources published over the last 5 years when analyzing the current state of research in the area under consideration, since this area of knowledge is rapidly developing.

Author Response

Responses to Reviewer 1’s comments

Dear reviewers:

On behalf of all co-authors, I’d like to express our sincere gratitude to you for your time and helpful comments / suggestions on the earlier version of our manuscript (Manuscript ID: ijms-1855892) titled “Systematic analysis and functional characterization of R2R3-MYB genes in Scutellaria baicalensis Georgi”. Your comments are very objective, which is very helpful for the improvement of our manuscript. Along with the revised manuscript with the “Track Changes”, we have included a point-to-point response to your comments. We are also ready to further improve the manuscript if any extra comments are made. Detailed responses to reviewers are given below.

Thanks again for your patient help.

All the best,

Xiaoyan Cao

Reviewer 1: Comments and Suggestions for Authors:

Dear authors.

Many plants contain metabolites with biological activities. The development of bioinformatics methods has opened up prospects for determining the mechanisms of formation of the properties and characteristics of metabolites. The scientific content of the manuscript justifies its publication, but some additions and modifications will significantly improve the quality of the article.

Major comments:

Point 1. In Introduction, the goal should be formulated.

Response 1: We appreciate the comments. Following your comments and suggestions, we rewritten the Introduction section and formulated the objective. Please refer to Line 98-101 for checking in the revised manuscript.

Point 2. L. 307 and in the text, at the beginning of the sentence the full biological name should be indicated without an abbreviation.

Response 2: Thank you very much for your comment. We have indicated in the revised version the full biological name "Scutellaria baicalensis" at the beginning of the sentence in the Discussion section (Line 309 in the revised manuscript).

Point 3. L.337, spacing should be.

Response 3: Thank you very much for your comment. We have made changes in the revised manuscript (Line 339 in the revised manuscript).

Point 4. All author assumptions about the functionality of the R2R3-Bmp gene require proof. This should be discussed in conclusion.

Response 4: Thank you very much for the reminder. Based on your comments, we have added corresponding contents in the Conclusion section. Please refer to Line 510-511 for checking in the revised manuscript.

Point 5. In the References, 47% of publications refer to 2017-2021 (the last 5 years); the remaining 53% of used sources are older than 5 years. It is recommended to increase the share of references to sources published over the last 5 years when analyzing the current state of research in the area under consideration, since this area of knowledge is rapidly developing.

Response 5: Thank you very much for your comment. We think it is very necessary to share the latest research papers in relevant fields, so we cited as many relevant references published in the last 5 years as possible, and replaced them in the revised manuscript.

Thanks again for your patient help.

Reviewer 2 Report

The authors show a complete analysis of the R2R3-SbMYBs gene family in S. baicalensis.

Minor comments

1.      The authors should further elaborate why SbMYB94 and SbMYB95 could not be assigned to any specific chromosome.

2.      While there are clades with representation of only R2-R3-SbMYBs and R2-R3-AthMYBs a particular function is not the only explanation, this can be discussed further in case the function of R2-R3MYBs  in Arabidopsis thaliana specific clades supports this explanation.

3.      Usually to unequivocally determine subcellular localization reporter genes are fused to both N-terminus and C terminus, can you elaborate why this was not done.

4.      The paper has several text editing mistakes:

·        The text has several typos that need attention. For example: The list of authors is incomplete, or the word “end” is in the wrong place.

·        Along the paper, several punctuation marks are wrong.

·        The discussion section has wordy paragraphs and some complex phases. Example below

These results indicated that gene duplications, in particular segmental duplications, may contribute to the expansion of R2R3-SbMYB genes, as the duplicated genes have similar gene structures and motif sequences”.

Author Response

Responses to Reviewer 2’s comments

Dear reviewers:

On behalf of all co-authors, I’d like to express our sincere gratitude to you for your time and helpful comments / suggestions on the earlier version of our manuscript (Manuscript ID: ijms-1855892) titled “Systematic analysis and functional characterization of R2R3-MYB genes in Scutellaria baicalensis Georgi”. Your comments are very objective, which is very helpful for the improvement of our manuscript. Along with the revised manuscript with the “Track Changes”, we have included a point-to-point response to your comments. We are also ready to further improve the manuscript if any extra comments are made. Detailed responses to reviewers are given below.

Thanks again for your patient help.

All the best,

Xiaoyan Cao

Reviewer 2: Comments and Suggestions for Authors:

The authors show a complete analysis of the R2R3-SbMYBs gene family in S. baicalensis.

Minor comments:

Point 1. The authors should further elaborate why SbMYB94 and SbMYB95 could not be assigned to any specific chromosome.

Response 1: Thank you very much for your comment. Due to the limitations of current plant genome sequencing and assembly technology, it is impossible to successfully mount all genes on specific chromosomes during the sequencing and assembly of S. baicalensis genome. Therefore, some genes can only be mounted on some contings or scarfords. Coincidentally, SbMYB94 and SbMYB95 can only be attached to conting391 and conting588, respectively, and these two contings are not attached to any specific chromosome. Therefore, SbMYB94 and SbMYB95 could not be assigned to any specific chromosome.

Point 2. While there are clades with representation of only R2R3-SbMYBs and R2R3-AtMYBs a particular function is not the only explanation, this can be discussed further in case the function of R2-R3MYBs in Arabidopsis thaliana specific clades supports this explanation.

Response 2: Thank you very much for your comment. Based on your comments, we reviewed the relevant literature and re-elaborated this part in the Discussion section. We can know that there are clades with representation of only R2R3-SbMYBs a particular function, which was either lost in Arabidopsis or acquired in S. baicalensis after divergence from the last common ancestor. The similar phenomena were observed in H. perforatum (Zhou W et al., 2020) and P. trichocarpa (Yang X et al., 2021). Furthermore, we taking Arabidopsis S12(no SbMYBs) as an example that the members of Arabidopsis S12 regulates glucosinolate synthesis. Glucosinolate is a secondary metabolite with nitrogen and sulfur, which is mainly present in Brassicaceae plants. In these plants, glucosinolate causes bitterness and pungency, and functions as an insecticide that protects from herbivores. Similar observations were found in sugar beet (Stracke et al., 2014), P. trichocarpa (Yang X et al., 2021) and P. bretschneideri (Li et al., 2016).

Point 3. Usually to unequivocally determine subcellular localization reporter genes are fused to both N-terminus and C terminus, can you elaborate why this was not done.

Response 3:Thank you very much for your comment. In this study, in order to further understand the potential function of R2R3 SbMYBs proteins, we predicted their subcellular localization and found that almost all of them localized in the nucleus via bioinformatics analysis. Subsequently, based on the existing subcellular expression vector "HBT-GFP-NOS vector" in our laboratory, we amplified the coding regions of six selected SbMYBs genes and fused them to the C-terminus of this vector for subcellular localization experiments. The results showed that all the six genes were indeed localized on the nucleus, which was completely consistent with our prediction. Therefore, we believe that it is not necessary to fuse these genes in the N-terminal for experiments.

Point 4. The paper has several text editing mistakes:

  • The text has several typos that need attention. For example: The list of authors is incomplete, or the word “and” is in the wrong place.
  • Along the paper, several punctuation marks are wrong.
  • The discussion section has wordy paragraphs and some complex phases. Example below

These results indicated that gene duplications, in particular segmental duplications, may contribute to the expansion of R2R3-SbMYB genes, as the duplicated genes have similar gene structures and motif sequences”.

Response 4: I am really sorry for the inconvenience and thank you for pointing out the problem. We have carefully checked the full text and carefully corrected text editing errors, making the following revisions to the manuscript:

  • Thank you very much for your suggestion, we checked and found that the word "and" was misplaced in the author list in the original manuscript, we have revised it in the revised version(line 5 in the revised manuscript). Meanwhile, we have also checked and revised other typos in the full text that need attention. For example: Substitute Z. mays for Z. may, and Nicotiana attenuata for Nicotiana attenuat.
  • I am very sorry for my negligence. We have checked the full text and made corrections carefully in the revised manuscript.
  • Thank you for raising this point. Based on your comments, we have checked and rewritten wordy paragraphs and some complex phases in the Discussion section, such as "These results indicated that gene duplications, especially segmental duplications, may contribute to the expansion of R2R3-SbMYB genes".

Thanks again for your patient help.

Reviewer 3 Report

The manuscript deals with analyzing the MYB-family genes in an important Chinese medicinal plant Scutellaria baicalensis Georgi. While studied extensively in other plants this gene family was scarcely characterized before in relation to the production of secondary metabolites with presumed medicinal effects. The manuscript presents the analysis of the largest known group of MYB-family genes in medicinal plants - the R2R3-MYB group. In this respect, it can serve as a basis for more detailed studies on the specific actions of various representatives of this subfamily on the production of medically-relevant secondary metabolites.

Being a specific study on the R2R3-MYB group in a particular plant the manuscript resorts to the typical methodology, used in other studies with a similar objective. This sometimes leads to presenting results and conclusions that could be omitted as way too obvious, i.e. "... results indicated that S. baicalensis and dicots presented more homologous gene pairs than S. baicalensis and monocots" - Lines 184-185. The overall impression from the present study is that it is more on the confirmative side, than describing some important new information on the studied subject. As such, it is of interest mainly to the specific group of researchers, dealing with the characterization of the secondary metabolite production in S. baicalensis.

Authors have a lax attitude to the spelling of Latin names of several species (i.e. Nicotiana attenuat instead of Nicotiana attenuata, Z. may instead of Z. mays, etc.). This, however, is unacceptable and should be corrected.

Author Response

Responses to Reviewer 3’s comments

Dear reviewers:

On behalf of all co-authors, I’d like to express our sincere gratitude to you for your time and helpful comments / suggestions on the earlier version of our manuscript (Manuscript ID: ijms-1855892) titled “Systematic analysis and functional characterization of R2R3-MYB genes in Scutellaria baicalensis Georgi”. Your comments are very objective, which is very helpful for the improvement of our manuscript. Along with the revised manuscript with the “Track Changes”, we have included a point-to-point response to your comments. We are also ready to further improve the manuscript if any extra comments are made. Detailed responses to reviewers are given below.

Thanks again for your patient help.

All the best,

Xiaoyan Cao

Reviewer 3: Comments and Suggestions for Authors:

The manuscript deals with analyzing the MYB-family genes in an important Chinese medicinal plant Scutellaria baicalensis Georgi. While studied extensively in other plants this gene family was scarcely characterized before in relation to the production of secondary metabolites with presumed medicinal effects. The manuscript presents the analysis of the largest known group of MYB-family genes in medicinal plants - the R2R3-MYB group. In this respect, it can serve as a basis for more detailed studies on the specific actions of various representatives of this subfamily on the production of medically-relevant secondary metabolites.

Being a specific study on the R2R3-MYB group in a particular plant the manuscript resorts to the typical methodology, used in other studies with a similar objective. This sometimes leads to presenting results and conclusions that could be omitted as way too obvious, i.e. "... results indicated that S. baicalensis and dicots presented more homologous gene pairs than S. baicalensis and monocots" - Lines 184-185. The overall impression from the present study is that it is more on the confirmative side, than describing some important new information on the studied subject. As such, it is of interest mainly to the specific group of researchers, dealing with the characterization of the secondary metabolite production in S. baicalensis.

Authors have a lax attitude to the spelling of Latin names of several species (i.e. Nicotiana attenuat instead of Nicotiana attenuata, Z. may instead of Z. mays, etc.). This, however, is unacceptable and should be corrected.

Response: I am really sorry for the inconvenience and thank you for pointing out the problem. Based on your comments, we carefully rechecked the full text and corrected the spelling errors of Latin names of several species carefully in the revised manuscript. Scutellaria baicalensis is a large-scale Chinese medicine that widely cultivated as crop in China. The main active ingredients of S. baicalensis are flavonoids, which possess several vital pharmacological antioxidation, anti-tumor, and anti-virus attributes. What is even more noteworthy is that it has significant influences on the treatment of COVID-19. Due to the great medicinal and economic value of S. baicalensis, it has been widely developed and utilized in the world. R2R3-MYB TFs play important roles in multitudinous significant physiological and biochemical processes, including hormone synthesis and signal transduction, response to biotic and abiotic stresses, and primary and secondary metabolism especially those affect nutritional and medicinal active ingredients or morphological and qualitative characteristics. However, the information of R2R3-MYB genes in S. baicalensis is lacking, which hinders research into R2R3-SbMYBs functionality to a certain extent. Hence, we performed a systematic bioinformatics analysis of R2R3-SbMYB gene family to understand the R2R3-MYB TF gene family's function, evolution, and expression profiles in S. baicalensis. In summary, these results provided novel insights into the functionality of R2R3-SbMYB, and established a foundation for further investigation of their biological function.

Thanks again for your patient help.
